# The Assessment of Psychosocial Work Conditions and Their Relationship to Well-Being: A Multi-Study Report

**DOI:** 10.3390/ijerph17051654

**Published:** 2020-03-04

**Authors:** Isabell Kuczynski, Martin Mädler, Yacine Taibi, Jessica Lang

**Affiliations:** Teaching and Research Area for Occupational Health Psychology, RWTH Aachen University, 52074 Aachen, Germany; mmaedler@ukaachen.de (M.M.); yacine.taibi@rwth-aachen.de (Y.T.); jlang@ukaachen.de (J.L.)

**Keywords:** well-being, psychosocial risk factors at work, questionnaire validation

## Abstract

The aim of this multi-study report is to present a questionnaire that enables researchers and practitioners to assess and evaluate psychosocial risks related to well-being. In Study 1, we conducted a cross-sectional online-survey in 15 German companies from 2016 to 2017 to verify factor- and criterion-related validity. Data consisted of 1151 employee self-ratings. Exploratory and confirmatory factor analyses resulted in an eight-factor structure (CFI = 0.902, RMSEA = 0.058, and SRMR = 0.070). All scales held to excellent internal consistency values (α = 0.65–0.90) and were related significantly to well-being (*r* = 0.17–0.35, *p* < 0.001). A second, longitudinal study in 2018 showed satisfying convergent and discriminant validity (*N* = 293) to scales from KFZA and COPSOQ. Test-retest reliability (*N* = 73; α = 0.65–0.88, *p* < 0.05) was also good. The instrument provides incremental validity above existing instruments since it explains additional variance in well-being.

## 1. Introduction

“Currently, psychosocial risks are recognized as one of the biggest challenges for occupational safety and health (OSH), as they are able to cause serious deterioration in workers’ physical and mental health, leading to significant consequences for organizations and society” [1] (p. 2). According to the European Union Labour Force Survey ad hoc module 2007, 55.6 million workers (27.9%) were exposed to factors that can adversely affect mental well-being [2]. The most frequent reported risk was time pressure or overload of work. Consequently, many countries all over the world have established national recommendations on how to manage psychosocial risk [2,3]. Risk management is defined as the identification, assessment, and prioritization of risks [4].

In Germany, the national recommendation is based on the Act on the Implementation of Measures of Occupational Safety and Health to Encourage Improvements in the Safety and Health Protection of Workers at Work (Arbeitsschutzgesetz). Since its update in 2013, the Arbeitsschutzgesetz explicitly includes the assessment of psychosocial factors at work (cf. Section 5 assessment of the conditions of work, number 6). The reason for the actualization was steadily rising days of sickness absence (second most frequent reason) and early retirements (most frequent reason) due to mental illnesses. The Joint German Occupational Safety and Health Strategy (GDA) published a recommendation for assessing psychosocial risks at work in 2014 [5]. The recommendation is based on the Arbeitsschutzgesetz and exclusively relates to working conditions as the subject of assessment. The GDA categorizes psychosocial risks according to four different domains: (1) work content, (2) work organization, (3) social relations, and (4) environmental conditions. All domains contain constructs related to mental well-being. For instance, in a cross-sectional multi-level analysis, well-being appears to have many work-related predictors, especially those related to aspects of the job content, such as job variety, job control but also to aspects of social relations like social support [6]. A study of employees among 34 European countries revealed a large number of psychosocial work factors associated with well-being (e.g., quantitative and emotional demands, development opportunities, significance of work, role clarity, leadership, social support, work-life balance) [7]. Factors of work environment (e.g., noise, climate, and lighting) also relate to mental well-being. For instance, noise reduction goes along with reduced stress symptoms [8].

In order to identify and prevent relevant psychosocial risks at work, the use of valid and reliable instruments is of utmost importance [1]. In recent decades, many instruments have been developed (see Tabanelli and colleagues [9] for an overview on 33 instruments for the assessment of work-related psychosocial factors (26 questionnaires and seven observational procedures). Thus, most psychosocial risk assessments are based on survey data. Past research has already discussed that, when designing a questionnaire for the assessment of work demands, specific design criteria have to be met throughout survey development to derive fruitful information from self-reports [10]. Common criticism regards biased item wording, questions not solely based on facts, and confusing the assessment of objective work factors with individual feelings of strain. In fact, among the 26 questionnaires identified by Tabanelli and colleagues, only 14 (54%) focused on psychosocial factors. Nearly half of the rest also assessed coping or strain [9]. In addition, the majority of those questionnaires assessed work demands from a person-centered perspective, potentially increasing the subjectivity bias [11]. Therefore, we have developed a new instrument that assesses working conditions from a third-person perspective [12].

PsyHealth is an objective and reliable instrument to assess psychosocial work conditions at item level in true work-settings [12]. We would like to examine whether the instrument has psychometric properties for use in research (Study 1) and test if PsyHealth measures psychosocial work conditions reliably over time and if it provides incremental validity above existing instruments in explaining variance in employee well-being (Study 2). Thus, the overall aim of this multi-study report is to link psychosocial risk factors at work to well-being with a newly developed self-report questionnaire minimizing potential sources of bias in the association of the work demand–strain concepts. Doing so, we hope to foster research and practice in managing psychosocial risks and promote mental well-being in working life.


**Study 1**


The first study is about the psychometric validity of PsyHealth. First, the factor validity and reliability were analyzed. Second, criterion-related validity was tested. We computed the relationship between the work factors and well-being. We have taken well-being as a criterion because high-quality studies have already shown that work conditions are related to well-being [6,7,8]. For instance, in a two-wave panel study work conditions influence psychological well-being [13]. Based on the evidence from comparable studies, we assume that all factors are related to well-being. As we code every work factor as a resource, we expect positive associations.

## 2. Materials and Methods

### 2.1. Data Collection

Study 1 was a two-year cooperation project between the study center and a social accident insurance in Germany. The social accident insurance advertised the study in their membership magazine and invited OSH specialists in their member companies to enroll their company in the study. Participants were thus job-holders of those companies. Local Bachelor and PhD projects delivered additional data. We carried out data collection from October 2016 to October 2017. We collected data online with a self-programmed software. The invitation to the survey was sent by the study center or its cooperation partner via e-mail. The e-mail contained the link to the survey and an identical access code for every participant of the same company, assuring anonymity. Participation was voluntary and terminable at any time. We presented the new instrument at first. Subsequently, the validation measure followed. The assessment took 10 to 15 minutes to complete. Participants agreed to the terms of use and privacy notice prior to starting the assessment. We did not asses sex and age in order to satisfy participants’ anonymity concerns and to add practical credibility by focusing on the working conditions and not on the individual. Prior to the study, the operational data protection officers from the university and the respective company and staff representatives had approved this approach.

### 2.2. Measures

#### 2.2.1. Assessement of Psychosocial Work Conditions

A focus group of OSH specialists generated an initial item pool of 48 items in 2016 (see Appendix A for an item description). The item pool included the assessment of work activity characteristics recommended by the official national initiative (i.e., GDA) such as decision latitude, social relations, emotional demands, work organization and work environment and additional characteristics like development opportunities and work-life balance as detected as relevant for mental health in past research.

As item and response scale generation might trigger response styles, especially in heterogeneous samples, wording is of utmost importance when developing new instruments [14]. To ensure that the items of our new instrument explicitly link to the activity, we formulated all items as activity-related and not person-related. This is one of the main formal differences between the new instrument and existing ones. Consequently, we formulated each item in the third-person by initiating the items with “The activity requires to […]” or “Within the activity it is […]” and not, for example, with “My job requires me […]” or “The job gives me […]”. With this formulation, we met the legal requirement of a condition-related risk assessment. Furthermore, we focused on the work activity instead of the job, since we wanted to assess the actual conditions during activity performance and not the stereotypical associations of a certain job. Items are formulated in a both positive and negative direction to avoid the formulation of double negations and response styles.

As with the formulation of the items, we attached great importance to using value-neutral response categories. For this reason, we decided to use an easily understandable verbalized frequency scale. For use in risk assessments, a frequency scale also has advantages for practical interpretation, as risk management aims to eliminate risky exposures or at least to minimize the exposure frequency of risks. Due to the limited capacity for humans to process information in the working memory, the rule of thumb “seven plus/minus two” applies to the optimal number in response units in psychological research [15]. However, for surveys in the general population, fewer categories should be used [16]. Since the instrument is intended for use in companies with jobholders of the general population, we chose a four-point response scale ranging from 0 “at no time or some of the time” to 3 “most or all of the time”. The response scale was the same for each item, except for the items relating to emotional demands in study 1 were 0 which were operationalized as “at no time”. We thought that was necessary for result interpretation. Only activities without any emotional demands were assessed to be risk free. Since it was possible that the requirement in question did not occur at the work activity to be analyzed [17], the response category “Not applicable” was also offered.

#### 2.2.2. Assessment of Well-Being

For criterion-related validity, mental well-being over the last two weeks was assessed with the WHO-5 Well-being Index (α = 0.87). The five-item questionnaire consists of questions that ask for the subjective well-being of the respondents. “The scale has adequate validity both as a screening tool for depression and as an outcome measure in clinical trials and has been applied successfully as a generic scale for well-being across a wide range of study fields” [18] (p. 174).

### 2.3. Participants

Overall data from 1435 participants of 15 companies were collected. The average response rate of completed PsyHealth questionnaires was 80.35%. Two respondents consistently indicated “not applicable”. We did not impute “not applicable” answers, since the participants were instructed to choose “not applicable” if the statement does not match a respondent’s activity. Accordingly, the data of the two participants contained no information that could be evaluated in the present context and were therefore excluded from further analysis. The final pilot sample consisted of 1151 employees. In order to prevent multiple sample use, we separated the data set into two subsets according to the requirements of the respective analysis method. For CFA, reliability and criterion-related analyses only data without any “Not applicable” answer were used (*n* = 564; data subset 2). We used the remaining data for EFA analyses (*n* = 587; data subset 1).

### 2.4. Statistical Analyses

To test the factor structure, we conducted exploratory factor analyses (EFA) using principal axis analysis with promax rotation. We applied parallel analysis using the online application of Gonzaga’s School of Business (https://analytics.gonzaga.edu/parallelengine/). Parallel analysis is more precise in extraction of the appropriate factor number, since the ‘eigenvalue greater than one criterion’ tends to overestimate the number of factors [19,20]. We selected strongly loading items for each scale (higher than 0.45). Since factors with less than three items are interpreted as weak and unstable, three items were the minimum number of items per scale [21]. To test the revised questionnaire we conducted a second EFA [22]. Thereafter, the exploratory extracted factors were verified by a CFA using IBM SPSS Amos 25 Graphics IBM Corp., Armonk, NY, USA). We evaluated the factor structure according to the recommendations from Hu and Bentler [23]. For the Maximum Likelihood Method, the authors emphasize the use of a two-index presentation. For Standardized Root-Mean-Square Residual (SRMR) values ≤0.08 represent good fit and for Root-Mean-Square Error of Approximation (RMSEA) values ≤0.06. For incremental fit indices, e.g., CFI values ≤0.95 are emphasized.

For the investigation of reliability, we computed internal consistency values (Cronbach’s alpha). We interpret consistency r < 0.60 as insufficient, 0.60 ≤ r < 0.70 as sufficient and r ≥ 0.70 as good reliability for group level studies according to the Committee On Test Affairs Netherlands (COTAN) [24].

In order to test criterion-validity, we conducted one-tailed Spearman’s rank correlations (*p* < 0.001 after Bonferroni correction).

## 3. Results

### 3.1. Factor Extraction

Exploratory factor analyses and parallel analyses extracted eight factors represented by 33 items (see Table 1). From the initial item pool of 48 items, 15 items had factor loadings beyond the threshold of 0.45. Those items were not included into the following CFA analyses. Although the chi-square value was significant (χ^2^ = 1337, df = 467, χ^2^/df = 2.86, *p* < 0.001), the other fit-indices (RMSEA = 0.058 and SRMR = 0.070) indicated good fit [24]. The CFI of 0.902 value is borderline, but acceptable due to the high item number (>10) [25]. The results indicate that the new instrument consists of a valid eight-factor structure of psychosocial work conditions.

### 3.2. Reliability

Table 1 and Table 2 contain the internal consistency values of the factors. All factors are reliable scales according to emphasized standard values in both subsets [24].

### 3.3. Criterion-Relatec Validity

Table 2 also contains the correlation coefficients between the work factors and well-being. All factors significantly related to well-being. We coded all psychosocial work factors in the same direction so that high values indicated good working conditions (i.e., resources). Thus, criterion-related validity was also confirmed. The strongest relationship with well-being was found for the scale assessing work intensity (r = 0.37, *p* < 0.001). This result is especially important as work intensity operationalized as time pressure or overload of work was the most frequent reported risk in the European Union Labour Force Survey 2007 [2].

## 4. Discussion

For use in research, instruments should meet the psychological measurement quality criteria of factor validity, scale reliability, and criterion-related validity. The current study delivers a suitable, reliable and valid factor model to describe the structure and quality of PsyHealth. Exploratory and confirmatory factor analyses indicate that the instrument consists of an eight-factor structure of psychosocial work conditions. One factor consists of eight items describing the work environment. Two distinguishable factors describe the social relations with colleagues and supervisors; each of them consists of four items. Another factor consists of five items describing work intensity. Four factors with three items each describe task clarity, decision latitude, work continuity, and emotional demands. It is important to note that our CFA model did not meet the cut-off value for the CFI. However, this might be due to the high number of items and the skewed data distribution [25]. Still, the indices of the eight-factor model are comparable to other instruments (e.g., The CFI of the Work Design Questionnaire (WDQ) was around 0.90 and interpreted as good fit) [26]. Since the eight-factor model covers a broad span of health-related work factors that are similar in content to the constructs of occupational stress theories, the new instrument might be suitable for future research in this context, especially since all factors measure reliably.

Not all items were suitable to be summarized into a reliable factor or respectively a psychological construct. The reason for that might be that more items are needed to better represent those constructs. This might be the case for the items describing aspects of task characteristics, qualification, information structure and work equipment. We decided not to add more items to achieve more constructs for the questionnaire as more items would have a negative impact on the economy of the instrument. Future research might analyze and expand the items to scales if necessary.

However, the factors evolved are most relevant for mental health as they consist of factors named by the Job-Demand-Control model [27], the Job-Demand-Control-Support model [28,29] and the Job-Demand-Resource model [30]. The assumption that these scales represent the constructs from the models satisfactorily needs to be validated in future research.

Criterion-related validity analyses of the identified eight scales show that all factors are related to well-being. The highest associations with well-being occurred for the work intensity scale. This indicates that a balanced ratio between work amount and time, beyond other factors, might be very important in the practical risk assessment. This result is line with comparable studies, which also find relationships between work intensity scales and well-being [7]. In addition, the items that have been removed from the research version of the questionnaire still significantly relate to well-being (see Appendix B). The only exception is skill utilization (item 10), which does not relate to well-being, but positive strain-balance (data not shown). Nevertheless, we decided to keep this item in the practitioners’ version of the questionnaire from the 2019th validation [12], as it seems to be a resource if present during task operation. One item that was removed from the practitioners’ version was the question of a fixed place of work (item 16) because it delivered contradictory results. The item relating to job security (item 17) also has been removed from the practitioners’ version. With the help of qualitative statements from interviewees (e.g., OSH experts), the response scale did not fit. The item required a yes/no answer rather than frequency. This might be the reason for the missing fit.

Results of Study 1 are cross-sectional. Therefore, the derived implications are limited to the description of associations. For the further evaluation of a questionnaire, it is recommended to use more than one reliability measure, since the variance of a test result can be influenced by many sources of error. Another important aspect for the evaluation in tests, is the examination of construct validity by the determination of the convergent and discriminant validity with already existing instruments that measure comparable constructs. Even more than convergent and discriminant validity, incremental validity is relevant in order to assess the added value of a new instrument.


**Study 2**


To test the stability of the assessment of psychosocial work conditions with PsyHealth, a longitudinal study design to measure test-retest reliability was conducted. The stability of the measurement of constructs depends on various aspects, such as memory and learning effects and changes to the environment. There are no strict standards given, concerning the optimum length of the test-retest interval [24]. We chose a two-to-four-week interval since we were interested in short-term mental health effects. This time interval should also be insensitive to serious changes in the work environment. Furthermore, we compared the strength of relationships between the scales from existing instruments and the new instrument with well-being. We analyzed convergent and discriminant validity using the multi-trait-multi-method approach (MTMM) [31].

## 5. Materials and Methods

### 5.1. Data Collection

For Study 2, participants were recruited during a student research project at the study center. We enrolled data collection from June 2018 to July 2018. Data collection occurred with the online survey platform SoSci Survey [32]. The survey was distributed by social media advertisement at two points in time, with the planned interval between the two measurement times being two to four weeks. For the second measurement, a personalized access code to the questionnaire was sent to the respondents 14 days after first participation via e-mail. The e-mail address was stored separately from the data. The data of the two measurement times of a participant were assigned to each other via an anonymous code. Participation was voluntary and terminable at any time.

### 5.2. Measures

We used two well-known validated instruments as comparative instruments, the Copenhagen Psychosocial Questionnaire (COPSOQ) [33] and the Short questionnaire for work analysis (in German: *Kurzfragebogen zur Arbeitsanalyse*, KFZA) [34]. To the best of our knowledge, these instruments are currently being used in occupational risk assessments in Germany. In addition, we have chosen these instruments because they were validated in Germany and therefore offer the best comparability. For economic reasons, we assessed the KFZA completely and only the scales from the COPSOQ that were not included in the KFZA but in PsyHealth, were added. The KFZA includes almost all constructs from the new instrument except task clarity and emotional challenges. Work environment is only operationalized by two items. Therefore, we used COPSOQ scales to assess this construct. Both existing instruments do not clearly distinguish between scales for relationship with supervisors and colleagues. We decided to use the KFZA scale assessing the relationship with supervisors and colleagues within one scale to compare it to both social relation scales of PsyHealth. Well-being was assessed again with the WHO-5 Well-being Index [18].

At time 1 (t1) the different instruments for measuring psychosocial work factors were randomly presented in order to prevent sequential effects. However, the order of the items was the same as in the validated version of the instruments. WHO-5 Well-being Index and the sociodemographic section had fixed points at the end of the questionnaire. At time 2 (t2) only the new instrument, WHO-5 Well-being questionnaire and the sociodemographic questions were presented.

### 5.3. Participants

At t1, 299 out of 404 respondents fully completed the questionnaire (74%). 116 respondents out of 122 participants completed the questionnaire at t2 (95%). Due to incorrect entries of the anonymized code, only 96 participants could be assigned to the data records of t1. The data set was adjusted for those participants who did not meet the following inclusion criteria: (a) more than 19 hours of work a week or (b) work in the same field of activity for more than one year. For the remaining *N* = 247 (t1) and *N* = 76 (t2) participants, the completion duration was considered, which was on average *M* = 13.70 (*SD* = 5.62) minutes for t1 and *M* = 6.03 (*SD* = 1.60) minutes for t2. To exclude inappropriate answers by not filling out conscientiously, data from participants, whose completion time at the first measurement time was less than one standard deviation from the mean, were not included. The final sample for statistical evaluation was *N* = 220 (t1) and *N* = 73 (t2).

### 5.4. Statistical Analyses

To verify test-retest-reliability, we computed one-tailed Spearman’s rank correlations (*p* < 0.05) between the two measurement times. As we interpret the retest-correlation coefficient as measure of reliability, we used the same cut-off values as for internal consistency in study 1. We also calculated Cronbach’s alpha for both measurement times. To test convergent and discriminant validity, we conducted one-tailed Spearman’s rank correlations (*p* < 0.05). We evaluated the MTMM matrix according to Campbell and Fiske [31]. We hypothesized that the eight scales of the new instrument will relate positively to the respective scale of existing instruments. There should be high and significant validity diagonals (hetero-method-mono-trait coefficients; convergent validity). Furthermore, we hypothesized that the convergent validity coefficients are higher than the correlations between different traits (discriminant validity). Finally, to test incremental validity, we performed hierarchical regression analyses in relation to well-being for the scales from the different instruments in order to identify if there was added value by the new instrument.

## 6. Results

One-tailed Spearman’s rank correlations of the eight scales of the new instrument between t1 and t2 indicate sufficient to good test-retest reliability (see Table 3). The scale assessing no emotional challenges results in the highest test-retest reliability. The scales measuring task clarity and work continuity are less stable, but still sufficient reliable. The reliability coefficient of the scale assessing work continuity was not sufficiently reliable at t1.

Table 4 presents the MTMM matrix for convergent and discriminant validity analyses. It contains the inter-correlations of each trait measured by each method. The validity diagonals were significantly different from zero and were sufficiently large. The new instrument satisfied the convergent validation criterion. The highest convergent correlation values occurred between the scales assessing emotional challenges and decision latitude. The lowest value was the one for task clarity. The reason for that might by differences at item level. For instance, the scale of the new instrument also includes an item on clear authority whereas the COPSOQ items focus on clear work orders and responsibility.

Discriminant validity has also been proven since the validity diagonal values were higher than the correlations between different traits. The only exception was task clarity. The convergent validity coefficient for task clarity was as high as the correlation coefficient between the task clarity scale assessed with the new instrument and the social relation scale assessed with the KFZA. Since the convergent value was generally not very high, we would not overinterpret the result, but we assume that through the addition of the item relating to authority, social aspects of role clarity were also assessed within the scale of the new instrument.

Finally, as shown in Table 4, all working factors, independent of instruments, correlated significantly with well-being. Though, emotional demands assessed with the COSPOQ were only marginally significant. This finding is especially important, because the emotional demands are a relevant factor in our increasingly service-based economy [35]. For this reason instruments should measure this construct sensitively.

Hierarchical regression analyses reveal that the new instrument explained significantly more variance in well-being than existing instruments (see Table A2, Table A3, Table A4, Table A5, Table A6, Table A7, Table A8 and Table A9 in Appendix C). For example, balanced work quantity assessed with the KFZA is a significant predictor (β = 0.44, *p* < 0.001), but if we added the work intensity scale of the new instrument to the analyses, this model explained 5% more variance (see Table A5 in Appendix C). The reason for this may be that the new instrument captures the construct with more items and thus represents it more extensively. In total, the scales of the new instrument explained 28% more variance in well-being than existing instruments, whereas the scales of the new instrument assessing work intensity, task clarity and no emotional challenges generated the strongest benefit in comparison to the scales of existing instruments. As these factors are identified as relevant in a cross-national study [7], they should be assessed as accurately as possible.

## 7. Discussion

Test-retest reliability coefficients indicate scale reliability over time. The scales assessing task clarity and work continuity are less stable than other scales. We suppose that both constructs might be “enacted” work characteristics that are more likely to be influenced by daily fluctuations [36]. Future research might help to differentiate latent job characteristics from enacted job characteristics by assessing work characteristics and their relationships with well-being in diary studies. It might be criticized that the sample size at t2 is small. Nevertheless, it should be noted that the relationships are significant despite this small sample size.

In relation to internal consistency, the scale assessing work continuity was not sufficiently reliable at t1, which might also be a reason for the low re-test reliability. Since the scale was reliable in study 1 and in t2 of study 2, we conclude that it is sufficiently reliable. The scale assessing no emotional challenges increased in internal consistency from study 1 to study 2. For this reason, we decided to use the same answer scale for every item for future use of the questionnaire.

According to the multi-trait-multi-method analyses, we conclude that the new instrument measures psychosocial work characteristics in convergence with existing instruments. Results on discriminant validity indicate that the scales largely assess distinct constructs. The reason for the relatively low discriminant validity values might be that all instruments are self-rating instruments assessing work characteristics. The most important finding from the comparisons of the instruments is that the new instrument explains more variance in well-being than existing ones.

## 8. Main Conclusions from Both Studies

The aim of this multi-study report is to test a new questionnaire for psychosocial risk assessment psychometrically. As a first step, we present a questionnaire with 33 items that measures psychosocial factors at work that are relevant for employees’ mental well-being. Moreover, the study results show that the new instrument explains more variance in well-being than existing ones. We emphasize that the added value of the new instrument is not only that it shows higher correlations to well-being, but also that measures for changing working conditions can be derived more accurately according to the condition-related item formulation. We therefore conclude that the new instrument makes a significant contribution to explaining the well-being of employees related to work conditions. The new instrument could therefore add value in work-related mental health research and practice.

One major limitation of both studies is that we assessed the working conditions and well-being with self-ratings. For this reason, we have to take the common-method bias [37] into account, when interpreting the present results. Future research should focus on validating the new instruments in relation to more objective work-related health outcomes. However, a previous study has shown that the subjective outcomes correspond to observer-ratings [12] and thus could be regarded as a rather “objective” assessment of the working conditions. Regarding objective health outcomes, physiological variables such as hair cortisol as a biomarker for chronic stress [38] or organizational outcome variables (e.g., sick leave days) could be considered.

For practical application, instruments should be time-efficient and results should be easy to interpret. Since items of the new instrument are formulated as condition-related, primary prevention measures can be derived easily. However, one might argue that only those working conditions that are perceived by the individual are relevant for health. The present study shows that no personal formulation is necessary to establish work demand–strain relationships. On the contrary, the correlations are higher for PsyHealth as activity-related instrument compared to person-related instruments. This could be due to the fact that inter-individual differences (e.g., due to personality) are reduced by the formulation of items in the third-person perspective [39]. Nevertheless, individual influences on the answers cannot be excluded.

Obviously, this result is limited to the comparison with instruments used in Germany. Nevertheless, advantages are that the new instrument is time-efficient as the implementation takes less than ten minutes. The instrument has been validated to contain scientifically deduced constructs. For result interpretation, all items are coded in one direction, so that general rules can be applied (i.e., the higher the value the better the working condition). We have opted for this kind of resource-oriented feedback to match the WHO’s orientation to define health as “a state of complete physical, mental and social well-being and not merely the absence of disease or infirmity” [40] (p. 1). This has the positive effect that mindsets are automatically resource-oriented and not deficit-oriented. In our experience, this facilitates cooperation with stakeholders in OSH management and facilitates the derivation of preventive measures.

PsyHealth assesses psychosocial risks at the micro level, i.e., the shared perception of specific working conditions by employees. Other questionnaires record psychosocial risks at the macro level, e.g., the Psychosocial Safety Climate (PSC) Scale that measures the common perception of “policies, practices and procedures for the protection of worker psychological health and safety” and describes PSC as “upstream organizational resource influenced largely by senior management” [41] (p. 579). The added value of the new instrument results in enabling the management to identify which specific working conditions pose a risk at all. This transparent presentation of the concrete risk may sensitize the management to increase PSC. In addition, PsyHealth could be used to assess the effectiveness of OSH initiatives (and ultimately of PSC). Future research could investigate if a joint assessment of both constructs establishes an effective and sustainable management of psychosocial risks at work. This approach would add value to the efforts of legislative and policy initiatives in Germany and other parts of the world. Despite all limitations, the PsyHealth questionnaire might be a promising alternative compared to existing survey assessment methods at the micro level. One of the advantages of this study is that we validated the questionnaire in an applied occupational setting (Study 1, 2) and in the real context of a psychosocial risk assessment (Study 1). Further analyses (e.g., on objective stress responses) are still required. The added value of the new instrument is that it explains well-being incrementally over existing instruments. The identification of the eight factors might foster bias-reduced research on work-related mental health. By presenting a new instrument for psychosocial risk assessment, fulfilling scientific (i.e., measurement quality) legal (i.e., task-related assessment) and practical (i.e., easy interpretation) requirements, the present work announces a potential solution to managing psychosocial risks in primary prevention and enhancing well-being at work.

## Figures and Tables

**Table 1 ijerph-17-01654-t001:** Results of the Exploratory factor analyses.

		Factor Loading								α
Factor	Items	1	2	3	4	5	6	7	8	
Work environ-ment	No unpleasant odors	**0.87**	0.05	−0.07	−0.03	0.03	−0.01	−0.13	0.07	0.88
	No heavy physical demands	**0.80**	−0.02	0.03	−0.01	0.02	−0.08	0.04	−0.03	
	No hazardous/biological agents	**0.74**	0.04	0.09	−0.01	−0.12	−0.01	−0.02	−0.08	
	Pleasant climate	**0.73**	−0.08	0.01	−0.07	0.00	0.05	0.06	0.07	
	Appropriate lighting	**0.72**	−0.02	0.05	−0.01	0.02	0.04	−0.02	−0.02	
	Quiet working environment	**0.67**	−0.08	0.03	0.08	0.02	−0.01	0.07	−0.10	
	Sufficient space	**0.67**	0.11	−0.14	0.07	0.06	−0.05	−0.10	0.01	
	Varied postures	**0.46**	0.04	−0.12	0.07	0.03	0.13	0.06	−0.03	
Social relations, colleagues	Coordination of joint tasks	0.03	**0.83**	0.00	−0.02	0.02	0.07	−0.04	−0.04	0.89
	Support among colleagues	−0.04	**0.83**	0.02	0.04	−0.04	−0.03	0.04	0.05	
	Professional conflict solving	0.00	**0.77**	0.10	−0.04	−0.03	0.05	0.01	0.02	
	Respect among colleagues	0.06	**0.74**	0.02	0.00	0.01	−0.07	0.04	0.01	
Social relations, supervisors	Acknowledgement from supervisor	−0.03	−0.05	**0.84**	0.02	0.01	−0.04	0.10	−0.01	0.90
	Helpful feedback from supervisor	0.07	0.07	**0.83**	0.02	−0.06	−0.01	−0.05	−0.02	
	Respect from supervisor	−0.09	0.08	**0.79**	0.06	0.06	0.03	−0.02	−0.01	
	Support from supervisor as needed	−0.04	0.07	**0.78**	−0.03	0.12	0.03	−0.07	−0.02	
Work intensity	Compliance with working hours	0.01	−0.01	−0.06	**0.85**	0.01	−0.02	0.05	−0.05	0.77
	Suitable ratio amount vs. time	0.05	−0.02	0.04	**0.71**	0.07	0.06	0.04	−0.02	
	Regular recovery breaks	−0.06	0.01	−0.04	**0.57**	0.00	0.01	−0.04	0.06	
	Time for core tasks	0.05	−0.07	0.12	**0.50**	−0.12	0.00	0.03	0.09	
	No changes in working hours	0.06	0.07	0.10	**0.47**	−0.01	−0.09	−0.09	0.01	
Task clarity	Clearly assigned responsibilities	0.07	−0.01	0.06	−0.05	**0.89**	0.03	0.02	−0.07	0.76
	Unambiguous work orders	0.00	−0.09	0.05	0.00	**0.78**	−0.01	−0.04	0.07	
	Authority for those responsible	−0.03	0.09	0.01	0.05	**0.51**	−0.07	0.10	0.08	
Work continuity	No interruptions (from people)	0.08	−0.07	0.00	−0.04	0.02	**0.82**	0.06	0.12	0.73
	No interruptions (due to ICT)	−0.04	0.12	−0.03	−0.10	−0.02	**0.72**	0.00	−0.12	
	No multiple tasks	−0.01	−0.05	0.05	.24	−0.05	**0.55**	−0.12	−0.03	
Decision latitude	Influence on task execution	0.08	−0.02	0.02	−0.02	−0.10	−0.06	**0.80**	0.02	0.66
	Influence on task content	−0.03	−0.01	0.05	−0.07	0.06	0.02	**0.71**	−0.10	
	Influence on task pace	−0.08	0.12	−0.17	.21	0.13	0.05	**0.47**	−0.02	
No Emotional challenges	No critical life events	0.01	−0.03	0.14	0.03	−0.06	0.01	−0.02	**0.77**	0.65
	No aggression/violence	−0.09	0.04	−0.18	0.04	0.09	−0.05	−0.16	**0.65**	
	No emotion suppression	0.15	0.04	0.02	−0.05	0.03	0.07	0.18	**0.48**	

Note. *N* = 587 (data subset 1); column numbers in bold belong to the same factor.

**Table 2 ijerph-17-01654-t002:** Results of the reliability and criterion-related validity analyses.

Work Factor	Cronbach’s α	Well-Being
Work environment	0.88	0.21 ***
Social Relations with colleagues	0.90	0.35 ***
Social Relations with supervisors	0.91	0.30 ***
Work intensity	0.80	0.37 ***
Task clarity	0.79	0.28 ***
Decision latitude	0.69	0.17 ***
Work continuity	0.74	0.30 ***
No emotional challenges	0.66	0.20 ***

Note. *N* = 564 (data subset 2); *** *p* ≤ 0.001.

**Table 3 ijerph-17-01654-t003:** Descriptive values and test-retest reliability.

Work Factor	t1 (*N* = 220)			t2 (*N* = 73)			
	*M*	*SD*	Reliability	*M*	*SD*	Reliability	Test-Retest Reliability
Work environment	2.13	0.61	0.79	2.14	0.65	0.83	0.79 ***
Social relations with colleagues	2.29	0.67	0.89	2.34	0.59	0.89	0.75 ***
Social relations with supervisors	1.85	0.79	0.86	1.83	0.75	0.88	0.84 ***
Work intensity	1.77	0.75	0.80	1.79	0.73	0.85	0.87 ***
Task clarity	1.98	0.66	0.78	2.08	0.64	0.76	0.68 ***
Decision latitude	1.11	0.65	0.68	1.19	0.69	0.72	0.72 ***
Work continuity	1.70	0.69	0.59	1.63	0.70	0.70	0.65 ***
No emotional challenges	1.71	1.01	0.82	1.71	1.04	0.84	0.88 ***

Note. Test-retest reliability coefficients result from one-tailed Spearman’s rank correlations (*** *p* ≤ 0.001); Reliability values are Cronbach’s alpha coefficients.

**Table 4 ijerph-17-01654-t004:** Multi-trait-multi-method matrix (*N* = 220).

	Factor	NI1	NI2	NI 3	NI 4	NI 5	NI 6	NI 7	NI 8	EI1	EI 2-3	EI 4	EI 5	EI 6	EI 7	EI 8
NI	1	1														
	2	0.32 ***	1													
	3	0.34 ***	0.51 ***	1												
	4	0.26 ***	0.35 ***	0.38 ***	1											
	5	0.19 **	0.42 ***	0.39 ***	0.35 ***	1										
	6	0.28 ***	0.27 ***	0.30 ***	0.23 ***	0.16 *	1									
	7	0.003	0.13	0.19 **	0.37 ***	0.20 **	0.05	1								
	8	0.17 *	0.12	0.22 ***	0.24 ***	0.051	0.01	0.18 **	1							
EI	1	**0.62 *****	0.30 ***	0.33 ***	0.26 ***	0.20 **	0.18 **	0.13 **	0.18 **	1						
	2–3	0.30 ***	**0.63 *****	**0.66 *****	0.35 ***	0.44 ***	0.29 ***	0.054	−0.01	0.26 ***	1					
	4	0.23 ***	0.26 ***	0.29 ***	**0.64 *****	0.25 ***	0.28 ***	0.42 ***	0.29 ***	0.32 ***	0.22 ***	1				
	5	0.16 *	0.35 ***	0.31 ***	0.14 **	**0.44 *****	0.12	0.03	0.07	0.21 ***	0.29 ***	0.13 *	1			
	6	0.33 ***	0.27 ***	0.45 ***	0.19 **	0.16 *	**0.70 *****	−0.02	0.07	0.20 ***	0.39 ***	0.16 *	0.24 ***	1		
	7	0.27 ***	0.32 ***	0.34 ***	0.45 ***	0.35 ***	0.19 **	**0.50 *****	0.30 ***	0.37 ***	0.27 ***	0.60 ***	0.17 *	0.15 *	1	
	8	0.03	−0.02	0.07	0.19 **	−0.04	−0.07	0.11	**0.78 *****	0.07	−0.11	0.22 ***	−0.004	−0.43	0.01	1
Well-being		0.28 ***	0.30 ***	0.36 ***	0.46 ***	0.24 ***	0.25 ***	0.23 ***	0.14 *	0.24 ***	0.33 ***	0.45 ***	0.16 *	0.26 ***	0.29 ***	0.12 ^†^

Note. Column numbers in bold represent significant hetero-method-mono-trait coefficients (validity diagonals); Correlation coefficients: * *p* ≤ 0.05, ** *p* ≤ 0.01, *** *p* ≤ 0.001 (Cronbach’s alpha/Spearman Brown); NI: new instrument, EI: existing instruments; KFZA: Short questionnaire for work analysis (in German: Kurzfragebogen zur Arbeitsanalyse); COPSOQ: Copenhagen Psychosocial Questionnaire; NI: 1 Work environment; 2 Social relations with colleagues; 3 Social relations with supervisors; 4 Work intensity; 5 Task clarity; 6 Decision latitude; 7 Work continuity; 8 No emotional challenges; 1, KFZA: Work environment; 2–3, KFZA: Social Relations; 4, KFZA: Quantity; 5, COPSOQ: Role clarity; 6, KFZA: Decision latitude; 7, KFZA: Working continuity; 8, COPSOQ: Emotional demands.

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
