# Peer review of "The Assessment of Psychosocial Work Conditions and Their Relationship to Well-Being: A Multi-Study Report"

_ijerph, 2020, doi:10.3390/ijerph17051654_

Round 1
Reviewer 1 Report
The article is very interesting, the methodology solid and well described as well as the analysis developed. Minor reviews of the english may be realized.

Author Response
Response to Reviewer 1 Comments
Point 1: The article is very interesting, the methodology solid and well described as well as the analysis developed. Minor reviews of the english may be realized.
Response 1: Thank you for your appreciation of the work. With the help of careful rereading of the co-authors and the helpful suggestions from reviewer 2, we hope to have achieved a satisfying improvement of linguistic. Please see minor changes in the new manuscript as listed below:
Line 10 - ‘researchers’
Line 24 - direct citation added and completed “Currently, psychosocial […]” (p. 2).
Line 27/28 - numbers have changed into ’55,6’ and ’27,9’
Line 32 to 38 - paragraph added to demonstrate why and how the matter has become increasingly important in Germany.
Line 38 - ‘In Germany’ deleted as it is now mentioned at the beginning of the added paragraph. The sentence now starts with ‘The Joint …’
Line 40 - ‘is based’ added after ‘recommendation’ (Singular)
Line 40 - German Occupational Safety and Health Act is now referred to as the German word for it ‘Arbeitsschutzgesetz’ as introduced in the text
Line 41 - ‘relates’
Line 41 - ‘exemplarily’ deleted
Line 47 - study ‘of’ not ‘with’
Line 48 - sentence has been rephrased for easier reading
Line 56 - same as 48
Line 72 - a new paragraph is added before ‘PsyHealth’
Line 78 - Hyphen added
Line 85 - a comma is added after ‘study’
Line 99 - “ “ “ “ “ ‘company’
Line 103 - ‘for satisfying’ replaced with ‘in order to satisfy’ and last half of sentence changed to read ‘to add practical credibility by focusing...’
Line 110 - ‘characteristic’ (not changed)
Line 113 - ‘opportunities’ (spelling corrected)
Line 117 - ‘as’ after ‘items’ added
Line 119 - ‘perspective’ deleted
Line 124 - ‘a’ after ‘in’ added
Line 130 - sentences restructured for a better understanding
Line 132 - add ‘,which’ after “...two” (not changed)
Line 132 - ‘of’ replaced with ‘in’
Line 138 - ‘units’ added after response
Line 139 - ‘necessary’ (spelling corrected)
Line 135 - ‘meant to be’ replaced with ‘assessed to be’
Line 146 - direct citation added and completed
Line 152 - ‘for’ replaced with ‘in’
Line 164 - ‘fewer replaced with ‘less’
Line 164 - generally replaced with ‘interpreted as’
Line 175 - 'spearman rank’ changed to 'Spearman's rank'
Line 198 - Exploratory factor analyses (uniform typography)
Line 217 - the semicolon replaced with a comma
Line 218 - the comma deleted
Line 224 - ‘respective’ deleted
Line 226/227 - model (uniform typography)
Line 240 - questionnaire deleted (confusing)
Line 247 - comma added after test
Line 256 - ‘26’ replaced by ‘24’ (wrong reference)
Line 263 - ‘Master Thesis’ replaced by ‘student research project’
Line 274 - paragraph added in order to underline our motives for choosing German instruments
Line 283 - social relation(s) scales
Line 302 - ‘spearman’ is now 'Spearman's...'
Line 306 - same as 302
Line 315 - same as 302
Line 320 - ‘validity’ added
Line 325 - ‘leben’ replaced by ‘level
Line 326 - ‘includes’
Line 326 - ‘whereat’ changed to ‘whereas’
Line 326 - changed to ‘focus(s)’
Line 336 - ‘Whereas’ replaced with ‘Though’ and a sentence was added the end of the paragraph in Line 335 at in order to explain the aspect more precisely.
Line 338 - ‘increasingly’ instead of ‘increasing’
Line 338/339 - sentence added in order to describe the aspect more precisely
Line 349 - ‘accurately’ instead of ‘accurate’
Line 352 - Spearman's
Line 360 - ‘discussion’ in bold (Formatting)
Line 368 - ‘sufficiently’ instead of ‘sufficient’
Line 370 - same as 368
Line 397 - ‘time-efficient’ instead of time-economic ‘in conduction’ deleted
Line 398 - ‘as’ added after ‘formulated’
Line 409 - comma after ‘interpretation’
Line 416 - paragraph added in order to refer to the PSC model, the measure and the added value of the instrument to foster management commitment
Line 428 - ‘questionnaire’ added
Line 429 - ‘at the micro level’ added
Line 436 - ‘announces’ instead of ‘presents’
Line 679 - Reference corrected (ESENER 2012 instead of 2018)
Line 684 - Reference corrected (first and last name exchanged)
Line 694 - Reference corrected (Formatting)
Line 698 - Reference corrected (Formatting)
Line 706 - Reference corrected (Formatting and abbreviation)
Line 742 - Reference corrected (Formatting)
Line 731 - Reference corrected (Formatting)
Line 733 - Reference corrected (Formatting)
Line 743 - Reference corrected (Formatting)
Line 773 - Reference corrected (Formatting)
Line 790 - New reference added

Reviewer 2 Report
I read this paper with great interest. It was both substantial and easy to go through.
I have minor suggestions for editing
Line 10 (first line of Abstract) - should be ‘researchers’ (typo)
33 - add ‘are based’ after ‘recommendations’
35 - delete ‘exemplarily’ (not a word)
40 - should be a study ‘of’ not ‘with’
61 - a new paragraph is needed before ‘PsyHealth’. This one is too long....
75 - needs a comma after ‘study’
88 - “ “ “ “ ‘company’
92 - replace ‘for satisfying’ with ‘in order to satisfy’
- then change last half of sentence to read ‘to add practical credibility by focusing...’
101 - ‘characteristic’ (typo)
102 - ‘opportunities’ (typo)
106 - add ‘as’ after ‘items’
108 - delete ‘perspective’
113 - add ‘a’ after ‘in’
119 - add a comma after ‘processing’ and change ‘effects’ to ‘affect’. (No sure what the second ‘processes’ means)
120 - add ‘,which’ after “...two”
122 - replace ‘of’ with ‘in’
125 - ‘necessary’ (typo)
126 - replace ‘meant to be’ with ‘assessed to be’
140 - replace ‘for’ with ‘in’
163 - 'spearman rank’ should be changed to 'Spearman's rank'
205 - replace the semicolon with a comma
206 - delete the comma
207 - delete ‘respectively’ (confusing)
214 - second ‘Model’ (typo)
252 - unclear what ‘Master Thesis’ refers to....
286 - ‘spearman’ should be 'Spearman's...'
290 - same as 286
299 - ditto re: spearman
309 - add ‘s’ to ‘include’
310 - change ‘whereat’ to ‘whereas’
310 - typo - change to ‘focus’
319 - replace ‘Whereas’ with ‘Therefore’
321 - should be ‘increasingly’ not ‘increasing’
331 - it’s ‘accurately’ not ‘accurate’
349 - ‘sufficiently’ not ‘sufficient’
351- same as 349
378 - change phrase to ‘time-efficient’ and delete ‘in conduction’
379 - add ‘as’ after ‘formulated’
390 - needs a comma after ‘interpretation’
Author Response
Response to Reviewer 2 Comments
Point 1: I read this paper with great interest. It was both substantial and easy to go through. I have minor suggestions for editing.
Response 1: Thank you for your interest in the paper and helpful suggestions for editing. We hope to have improved grammar and reading fluency as suggested. We did not change the suggestions for line 110 and 132 as it would have changed the semantic of the sentence. We also edited some references. Please see minor changes in the new manuscript as listed below:
Line 10 - ‘researchers’
Line 24 - direct citation added and completed “Currently, psychosocial […]” (p. 2).
Line 27/28 - numbers have changed into ’55,6’ and ’27,9’
Line 32 to 38 - paragraph added to demonstrate why and how the matter has become increasingly important in Germany.
Line 38 - ‘In Germany’ deleted as it is now mentioned at the beginning of the added paragraph. The sentence now starts with ‘The Joint …’
Line 40 - ‘is based’ added after ‘recommendation’ (Singular)
Line 40 - German Occupational Safety and Health Act is now referred to as the German word for it ‘Arbeitsschutzgesetz’ as introduced in the text
Line 41 - ‘relates’
Line 41 - ‘exemplarily’ deleted
Line 47 - study ‘of’ not ‘with’
Line 48 - sentence has been rephrased for easier reading
Line 56 - same as 48
Line 72 - a new paragraph is added before ‘PsyHealth’
Line 78 - Hyphen added
Line 85 - a comma is added after ‘study’
Line 99 - “ “ “ “ “ ‘company’
Line 103 - ‘for satisfying’ replaced with ‘in order to satisfy’ and last half of sentence changed to read ‘to add practical credibility by focusing...’
Line 110 - ‘characteristic’ (not changed)
Line 113 - ‘opportunities’ (spelling corrected)
Line 117 - ‘as’ after ‘items’ added
Line 119 - ‘perspective’ deleted
Line 124 - ‘a’ after ‘in’ added
Line 130 - sentences restructured for a better understanding
Line 132 - add ‘,which’ after “...two” (not changed)
Line 132 - ‘of’ replaced with ‘in’
Line 138 - ‘units’ added after response
Line 139 - ‘necessary’ (spelling corrected)
Line 135 - ‘meant to be’ replaced with ‘assessed to be’
Line 146 - direct citation added and completed
Line 152 - ‘for’ replaced with ‘in’
Line 164 - ‘fewer replaced with ‘less’
Line 164 - generally replaced with ‘interpreted as’
Line 175 - 'spearman rank’ changed to 'Spearman's rank'
Line 198 - Exploratory factor analyses (uniform typography)
Line 217 - the semicolon replaced with a comma
Line 218 - the comma deleted
Line 224 - ‘respective’ deleted
Line 226/227 - model (uniform typography)
Line 240 - questionnaire deleted (confusing)
Line 247 - comma added after test
Line 256 - ‘26’ replaced by ‘24’ (wrong reference)
Line 263 - ‘Master Thesis’ replaced by ‘student research project’
Line 274 - paragraph added in order to underline our motives for choosing German instruments
Line 283 - social relation(s) scales
Line 302 - ‘spearman’ is now 'Spearman's...'
Line 306 - same as 302
Line 315 - same as 302
Line 320 - ‘validity’ added
Line 325 - ‘leben’ replaced by ‘level
Line 326 - ‘includes’
Line 326 - ‘whereat’ changed to ‘whereas’
Line 326 - changed to ‘focus(s)’
Line 336 - ‘Whereas’ replaced with ‘Though’ and a sentence was added the end of the paragraph in Line 335 at in order to explain the aspect more precisely.
Line 338 - ‘increasingly’ instead of ‘increasing’
Line 338/339 - sentence added in order to describe the aspect more precisely
Line 349 - ‘accurately’ instead of ‘accurate’
Line 352 - Spearman's
Line 360 - ‘discussion’ in bold (Formatting)
Line 368 - ‘sufficiently’ instead of ‘sufficient’
Line 370 - same as 368
Line 397 - ‘time-efficient’ instead of time-economic ‘in conduction’ deleted
Line 398 - ‘as’ added after ‘formulated’
Line 409 - comma after ‘interpretation’
Line 416 - paragraph added in order to refer to the PSC model, the measure and the added value of the instrument to foster management commitment
Line 428 - ‘questionnaire’ added
Line 429 - ‘at the micro level’ added
Line 436 - ‘announces’ instead of ‘presents’
Line 679 - Reference corrected (ESENER 2012 instead of 2018)
Line 684 - Reference corrected (first and last name exchanged)
Line 694 - Reference corrected (Formatting)
Line 698 - Reference corrected (Formatting)
Line 706 - Reference corrected (Formatting and abbreviation)
Line 742 - Reference corrected (Formatting)
Line 731 - Reference corrected (Formatting)
Line 733 - Reference corrected (Formatting)
Line 743 - Reference corrected (Formatting)
Line 773 - Reference corrected (Formatting)
Line 790 - New reference added

Reviewer 3 Report
Working conditions and wellbeing has been an important topic in the studies of public health. In this paper, the authors proposed and tested a new questionnaire for psychosocial risk assessment. They were able to provide sufficient results to support the validity of the new instrument and suggest how the new questionnaire may be a potential alternative to existing assessment methods, as it seems to provide better measurement quality, task-related assessment, and easy interpretation. The author also suggested some limitations of the study and how the instruments could possibly be further improved by assessing objective health outcomes. I personally look forward to the further development and future application of the PsyHealth in Germany and/or elsewhere in the world.
Suggestion only:
As some readers might not be familiar with the situations in Germany, it would be very helpful if the authors could introduce one or two examples of social issues related to work and well-being in Germany (suicide rate among workers?), and add some discussions as to how this subject matter has become increasingly important in Germany.
Author Response
Response to Reviewer 3 Comments
Point 1: As some readers might not be familiar with the situations in Germany, it would be very helpful if the authors could introduce one or two examples of social issues related to work and well-being in Germany (suicide rate among workers?), and add some discussions as to how this subject matter has become increasingly important in Germany.
Response 1: Many thanks for the constructive evaluation of the manuscript and the motivating words for the further development and use of the instrument. In relation to your suggestion, we now refer to the reasons that catalyzed the importance of mental health prevention in Germany in the introduction. We have limited the social aspects to work-related risks in order to set a focus here.
Please see page 2 line 32 to 38:
“In Germany, the national recommendation is based on the Act on the Implementation of Measures of Occupational Safety and Health to Encourage Improvements in the Safety and Health Protection of Workers at Work (Arbeitsschutzgesetz). Since its update in 2013, the Arbeitsschutzgesetz explicitly includes the assessment of psychosocial factors at work (cf. section 5 Assessment of the conditions of work, number 6). The reason for the actualization were steadily rising days of sickness absence (second most frequent reason) and early retirements (most frequent reason) due to mental illnesses.”

Reviewer 4 Report
The authors have clearly worked hard on this paper for which they are to be congratulated.
The subject of psychosocial factors in the workplace has gained a great deal of traction in recent years.
The work of Dolland and Bakker for example in developing the PsychoSocial Safety Climate model and associated PSC 12 measure has been validated in many studies. So it is a surprise that this work is nowhere referenced in the paper, particularly since the PSC 12 would appear to be the only modern scale apart from the new one they are promoting.
It would seem to me to be important for them acknowledge this work and to indicate what the benefits of their new scale are compared with the PSC 12.
In addition, the works on PSC undertaken to date show compellingly that PSC is essentially a 'top down' phenomenon, and indeed without full 'buy in' by management, any efforts to improve workplace climate is pointless.
None of this seems to be referenced or understood in the paper as it stands.
I really don't think it is adequate to develop a new measure in this important area without demonstrating value compared with existing and well validated scales.
It would be important to me as an assessor to see these matters addressed before obtaining my imprimateur.
Author Response
Response to Reviewer 4 Comments
Point 1: The authors have clearly worked hard on this paper for which they are to be congratulated. The subject of psychosocial factors in the workplace has gained a great deal of traction in recent years. The work of Dolland and Bakker for example in developing the PsychoSocial Safety Climate model and associated PSC 12 measure has been validated in many studies. So it is a surprise that this work is nowhere referenced in the paper, particularly since the PSC 12 would appear to be the only modern scale apart from the new one they are promoting.
Response 1: Thank you for your acknowledgement of the work required for this paper and your critical evaluation. We added a paragraph in the method section of Study 2 in order to underline our motives for choosing German instruments.
Please see page 10 line 274 to 276:
“To the best of our knowledge, these instruments are currently being used in occupational risk assessments in Germany. In addition, we have chosen these instruments because they were developed in German and therefore offer the best comparability.”
Point 2: It would seem to me to be important for them acknowledge this work and to indicate what the benefits of their new scale are compared with the PSC 12. In addition, the works on PSC undertaken to date show compellingly that PSC is essentially a 'top down' phenomenon, and indeed without full 'buy in' by management, any efforts to improve workplace climate is pointless.
Response 2: We appreciate the Psychosocial Safety Climate model and now refer to the model, the measure and the added value of the instrument to foster management commitment in the discussion section of our paper. In consequence, we added a new reference; Dollard & Bakker, 2010.
Please see page 15 line 416 to 427:
“PsyHealth assesses psychosocial risks at the micro level, i.e. the shared perception of specific working conditions by employees. Other questionnaires record psychosocial risks at the macro level, e.g. the Psychosocial Safety Climate (PSC) Scale that measures the common perception of “policies, practices and procedures for the protection of worker psychological health and safety” and describes PSC as “upstream organizational resource influenced largely by senior management” (Dollard & Bakker, 2010, p. 579). The added value of the new instrument results in enabling the management to identify which specific working conditions pose a risk at all. This transparent presentation of the concrete risk may sensitize the management to increase PSC. In addition, PsyHealth could be used to assess the effectiveness of OSH initiatives (and ultimately of PSC). Future research could investigate if a joint assessment of both constructs establishes an effective and sustainable management of psychosocial risks at work. This approach would add value to the efforts of legislative and policy initiatives in Germany and other parts of the world.”

Round 2
Reviewer 4 Report
The authors have responded to assessors comments and suggestions in a considered and thoughtful way, which has strengthened the significance of their paper.